# Video Laryngoscopic Intubation Using the King Vision^TM^ Laryngoscope in a Simulated Cervical Spine Trauma: A Comparison Between Non-Channeled and Channeled Disposable Blades

**DOI:** 10.3390/diagnostics10030139

**Published:** 2020-03-03

**Authors:** Jiri Votruba, Tomas Brozek, Jan Blaha, Tomas Henlin, Tomas Vymazal, Will Donaldson, Pavel Michalek

**Affiliations:** 1First Department of Tuberculosis and Respiratory Care, 1st Medical Faculty of the Charles University and General University Hospital, 128 00 Prague, Czech Republic; jiri.votruba@vfn.cz; 2Department of Anaesthesia and Intensive Medicine, 1st Medical Faculty of the Charles University and General University Hospital, 128 00 Prague, Czech Republic; tomas.brozek@vfn.cz (T.B.); jan.blaha@vfn.cz (J.B.); 3Medical Faculty, Masaryk University, 625 00 Brno, Czech Republic; 4Department of Anaesthesia and Intensive Medicine, 1st Medical Faculty of the Charles University and University Military Hospital, 128 00 Prague, Czech Republic; tomas.henlin@uvn.cz; 52 nd Medical Faculty of the Charles University and University Hospital Motol, 150 06 Prague, Czech Republic; tomas.vymazal@fnmotol.cz; 6Department of Anaesthesia, Antrim Area Hospital, Antrim BT41 2RL, UK; willdonaldson@mac.com

**Keywords:** videolaryngoscopy, King Vision™ laryngoscope, channeled blade, non-channeled blade, cervical spine injury

## Abstract

Videolaryngoscopes may reduce cervical spine movement during tracheal intubation in patients with neck trauma. This manikin study aimed to compare the performance of disposable non-channeled and channeled blades of the King Vision™ videolaryngoscope in simulated cervical spine injury. Fifty-eight anesthesiologists in training intubated the TruMan manikin with the neck immobilized using each blade in a randomized order. The primary outcome was the time needed for tracheal intubation, secondary aims included total success rate, the time required for visualization of the larynx, number of attempts, view of the vocal cords, and subjective assessment of both methods. Intubation time with the channeled blade was shorter, with a median time of 13 s (IQR 9–19) *vs*. 23 s (14.5–37.5), *p* < 0.001, while times to visualization of the larynx were similar in both groups (*p* = 0.54). Success rates were similar in both groups, but intubation with the non-channeled blade required more attempts (1.52 *vs*. 1.05; *p* < 0.001). The participants scored the intubation features of the channeled blade significantly higher, while visualization features were scored similarly in both groups. Both blades of the King Vision™ videolaryngoscope are reliable intubation devices in a simulated cervical spine injury in a manikin model when inserted by non-experienced operators. The channeled blade allowed faster intubation of the trachea.

## 1. Introduction

Videolaryngoscopy-guided tracheal intubation is gradually moving from operating rooms to emergency medicine departments and to the out-of-hospital setting [1]. Videolaryngoscopy may offer some advantages against Macintosh laryngoscopy, such as improved and easier visibility of the laryngeal inlet in normal and difficult laryngoscopy scenarios, less need of force during intubation attempts, and visualization of the intubation process on the monitor. Patients with potential cervical spine trauma have a higher incidence of difficult intubation due to neck immobility caused by inline stabilization or the application of a cervical collar, which may also lead to restricted mouth opening [2]. The use of videolaryngoscopes may reduce cervical spine movements [3] and improve the view of the vocal cords in patients with cervical spine injury [4]. Various videolaryngoscopes are currently available on the market. Non-channeled devices require insertion of a stylet into the tracheal tube prior to intubation or the use of a bougie, whereas channeled videolaryngoscopes possess a dedicated channel for tracheal tube insertion [5]. The first generation of King Vision^TM^ laryngoscope (King Systems, Noblesville, IN, USA) was invented in 2013 and incorporated into clinical practice in 2014 [6]. The most recent, second generation of the device (Ambu, Ballerup, Denmark) is characterized by the reusable body (containing the monitor and flexible optical stylet) and by its use of interchangeable disposable plastic channeled and non-channeled blades [7]. Recently, paramedic services in the United States have been trained using this device in prehospital tracheal intubation [8]. Several studies have compared the performance of King Vision™ videolaryngoscopes with standard blades, however, data comparing the performance of disposable non-channeled versus channeled videolaryngoscope blades are not available. The aim of this simulation study was to compare the performance of single-use non-channeled standard aBlade™ and channeled aBlade™ blades of the KingVision™ videolaryngoscope in a simulated cervical spine injury when inserted by relatively inexperienced operators. 

## 2. Materials and Methods 

### 2.1. Study Design 

The design of this simulation study was prospective, non-blinded, randomized, and cross-over. Study participants were anesthesiology and intensive medicine residents (years 1–5 in training) from four different teaching hospitals. The study protocol was reviewed and approved by the Ethical Committee of the General University Hospital (No. 89/15 S-IV VFN, 19 February 2015, Ethics Committee of the General University Hospital, Prague). All participants signed informed consent for their voluntary participation in the study. In total, 58 participants completed the study. One type of airway simulation manikin, TruMan (Trucorp Ltd., Belfast, UK) was used for all interventions. The participants had no previous experience with the King Vision^TM^ videolaryngoscope before commencement of the study, however, most of them were familiar with other types of videolaryngoscopes. Video demonstration of both blades was provided to the participants, as well as a brief live demonstration. Participants each performed one trial of intubation with each blade before the start of the study. 

### 2.2. Experimental Procedures

The manikin had its neck immobilized using a cervical collar (Figure 1), which led to a laryngeal view consistent with a Cormack–Lehane grade of 3 (only epiglottis visible) using a standard Macintosh laryngoscope blade. This was confirmed by two independent consultant anesthesiologists who were not participating in the study. The manikin was placed in the supine position, and no artificial light was used in the study room. Participants intubated the manikin with each blade in a randomized order. The two currently available disposable blades of the King Vision^TM^ videolaryngoscope were used in this study. The non-channeled standard aBlade^TM^ (Figure 2) has a maximum anteroposterior diameter of 11 mm, whilst the diameter of the channeled aBlade^TM^ (Figure 2) is 17 mm. Randomization codes were created by using randomization software (www.graphpad.com) and sealed envelopes. All participants initiated their scenario with bag-mask ventilation, then intubated the manikin, and finally connected the endotracheal tube to the self-inflatable bag and confirmed the position of the endotracheal tube with a stethoscope. Endotracheal tubes were inserted directly through the leading channel in the channeled blade, whereas for intubation with the non-channeled blade, a preformed semirigid stylet was used to guide the tube through the vocal cords. A maximum of three attempts for tracheal intubation was allowed. 

### 2.3. Study Outcomes

The primary outcome measure was the time required for tracheal intubation. This interval started when the participant first held the videolaryngoscope (this was in the standing position) and ended when the tracheal tube entered the trachea. An independent observer confirmed the correct position of the tube between the vocal cords on the monitor of the videolaryngoscope. We evaluated several secondary outcomes. The total success rate was recorded as well as the number of attempts. The time required for visualization of the larynx was measured from first holding the videolaryngoscope until the first view of the vocal cords was obtained. The total time required for initiation of artificial ventilation (“no oxygenation time“) was measured from first holding the videolaryngoscope until the first successful breath, as confirmed by chest movements. The quality of visualization of the vocal cords was recorded using the ”Percentage of Glottic Opening“ (POGO) score [9,10] and Cormack–Lehane grading [11] (Appendix A). 

The participants filled in a feedback questionnaire immediately after completion of the study. Subjective assessment was undertaken of the quality of the glottic view and comparative ease of intubation between the blades. The Likert scale (1–5 points; 1—very good, 2—good, 3—neutral, 4—poor, 5—very poor) was used for these assessments.

### 2.4. Statistics

Sample size calculation was performed prior to the commencement of the study. Based on the results of previous studies [6,12], we chose a time of 20 s for successful tracheal intubation, with a meaningful difference between the groups of 4 s (20%). With an alpha error of 0.05 and a beta error (power) of 0.8, the minimum sample size was calculated to be 49 participants. A total number of 58 participants was chosen to compensate for potential dropouts. The data was checked for a normal distribution (Shapiro–Wilk test) and subsequently analyzed using the non-parametric Mann–Whitney U test for continuous data or Fisher´s exact test for categorical data. Statistical software InStat (GraphPad Inc., La Jolla, CA, USA) was used for all comparisons. In this study, *p* values lower than 0.05 were considered statistically significant.

## 3. Results

All 58 physicians in training completed the study. Their demographic data are summarized in Table 1. The participants inserted tracheal tubes below the vocal cords significantly faster through the channeled aBlade—median 13 s (25–75 interquartile range (IQR) 9–19)—than with the non-channeled blade—median time of 23 s (IQR 14.5–37.5) (Table 2, Figure 3). Time to visualize the laryngeal inlet in both groups was close to 10 s and did not reach statistical difference between the groups. The total time between starting the process of tracheal intubation and initiation of artificial ventilation was significantly longer with the use of the non-channeled blade-median, 29 s *vs*. 20 s; *p* < 0.001. The total success rate of tracheal intubation was similar in both groups, while for tracheal intubation with the non-channeled blade, the total success rate did not differ between the blades, although intubation with non-channeled blade required significantly more attempts (1.52 *vs*. 1.05; *p* = 0.0007). There was one failure to intubate the manikin with the non-channeled blade. The participant was able to achieve quite a good visualization of the larynx but could not pass the endotracheal tube into the trachea. Laryngoscopic scores were similar in the groups using both POGO and Cormack–Lehane assessments. The ease of tracheal intubation was evaluated by the participants as significantly better for the channeled aBlade in comparison with the non-channeled aBlade, while visualization features were marked as similar in both groups (Table 3).

## 4. Discussion

Videolaryngoscopy blades may be divided into non-channeled blades and channeled blades. Each of the blades has its advantages and disadvantages [4,5]. Non-channeled blades are generally thinner, easier to insert, and provide a good quality view to the vocal cords even in the case of significantly limited mouth opening. Their main disadvantage is that insertion of the tracheal tube needs a special angulated introducer and there may be a risk of trauma to the oropharyngeal soft tissues caused by blind spots during tracheal tube insertion. The channeled blades provide a reliable direct guide for tracheal tube insertion but may be more difficult to introduce in the case of limited mouth opening, intraoral swelling, or bulky tongue. 

The results of this study suggest that, in a simulated scenario, both blades of the King Vision™ videolaryngoscope may be suitable devices for airway management in cervical spine trauma. The success rates of both devices were very high and did not differ between the groups. However, tracheal intubation was faster when using the channeled blade of the device. There was no difference in the time of visualization of the vocal cords between the blades, but there was a significant difference in duration of actual tracheal tube insertion into the trachea. This fact suggests that manipulation with the tracheal tube inside the oral cavity may be significantly more difficult when there is no guiding channel for the tracheal tube. This finding is in agreement with the subjective assessment of the performance of both blades from the participants. They did not report any difference between the blades in terms of visualization of the laryngeal inlet but found the insertion of the endotracheal tube significantly more difficult for the blade without a guiding channel. 

Several studies have assessed the performance of King Vision™ laryngoscopes in both experimental and clinical settings. In one study, the reusable channeled and non-channeled blades of the first generation of King Vision™ laryngoscope were compared with the Macintosh laryngoscope blade for intubation of the manikin trachea by inexperienced operators [5]. The authors found that only the channeled blade showed a similar performance to the Macintosh laryngoscope, whilst intubation with non-channeled blades was associated with more failures and prolonged intubation times. Murphy et al. compared the channeled King Vision™ reusable blades with the Macintosh laryngoscope in simulated manikin and cadaver difficult airways [13]. Tracheal intubation with the King Vision™ was faster and more successful in cadavers than with the classical Macintosh laryngoscope. The King Vision™ videolaryngoscope was found to be equally as effective when intubating patients with suspected cervical spine injury (who had immobilization with a cervical collar) as the C-MAC or D-MAC laryngoscopes [14]. However, the authors did not report whether a channeled or non-channeled blade was used. The channeled King Vision™ blade was also compared with the Macintosh laryngoscopy in terms of cervical spine motions during tracheal intubation in patients with normal cervical spine [15] and was found to be superior to the conventional technique. The use of a channeled disposable blade shortened intubation time and improved the success rate in comparison with the Macintosh laryngoscope in patients with cervical spine immobilization and without predictors of airway difficulty [16]. No studies comparing the performance of disposable channeled and non-channeled blades in cervical spine immobilization have been published so far. 

The main limitation of this study was that it was performed on simulators and not on human subjects [17]. However, the findings of this manikin trial may serve as a starting point for randomized trials performed on patients under anesthesia with a simulated cervical spine injury or on patients with actual suspected cervical spine trauma. The portable and user-friendly design of the King Vision™ laryngoscope may predispose this device for use mainly in critical airway scenarios in the prehospital emergency setting [18,19].

## 5. Conclusions

Both disposable blades of the KingVision™ videolaryngoscope were reliable intubation devices in a simulated cervical spine injury using an airway manikin when inserted by non-experienced operators. However, the participants placed tubes into the trachea significantly faster with the channeled blade and also rated it as being better than the non-channeled blade. This may favor channeled blades in this type of pre-hospital intubation scenario.

## Figures and Tables

**Figure 1 diagnostics-10-00139-f001:**
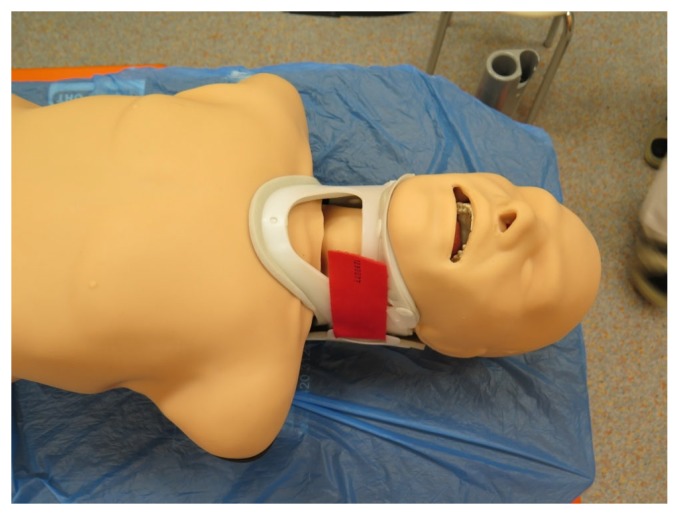
Truman manikin with the neck immobilized using the cervical collar.

**Figure 2 diagnostics-10-00139-f002:**
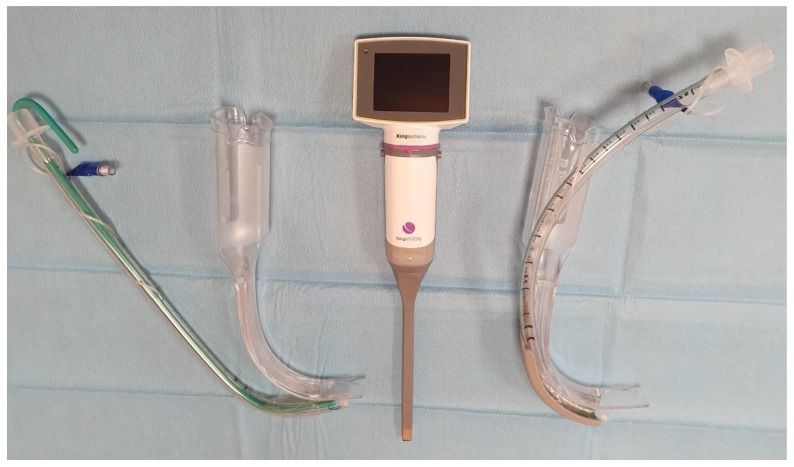
Interchangeable channeled (right) and non-channeled (left) blades of the King Vision^TM^ video laryngoscope. A tracheal tube is preloaded to the guiding channel of the channeled blade, while for tracheal intubation with the non-channeled blade, a curved stylet is inserted into the tube and the vocal cords are approached from the lateral side of the blade.

**Figure 3 diagnostics-10-00139-f003:**
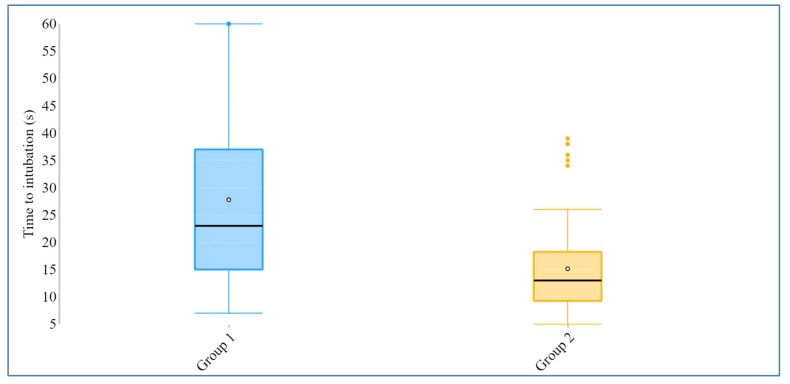
Box plot of the primary outcome—time to intubation. Group 1: non-channeled blade; group 2: channeled blade.

**Table 1 diagnostics-10-00139-t001:** Demographic data of the participants.

	*N*	%
Gender		
Males	26	55
Females	32	45
Level of training		
Less than 1 year	13	22
1–2 years	12	21
2–3 years	18	31
3–5 years	11	19
More than 5 years	4	7
Experience in videolaryngoscopy		
None	5	9
1–10 intubations	27	46
11–50 intubations	19	37
51–100 intubations	4	7
More than 100 intubations	3	5

**Table 2 diagnostics-10-00139-t002:** Primary and secondary outcomes of the study.

Parameter	Non-Channeled Blade	Channeled Blade	*p*
Time to intubation (s)	23 (7–91) [14.5–37.5]	13 (5–39) [9–19]	<0.001 *
Time to visualization (s)	8 (3–26) [5.75–11]	9 (4–36) [6–12]	0.54
Time to ventilation (s)	29 (13–99) [21.5–45]	20 (12–46) [16.75–26]	<0.001 *
Total success rate (%)	98.3	100	1.00
Number of attempts	1 (1–4) [1–2]	1 (1–2) [1]	<0.001 *
/Mean ± SD/	/1.52 ± 0.7/	/1.05 ± 0.22/	
Percentage of glottic opening	2 (1–5) [2–3]	2 (1–4)[1.75–2]	0.055
Cormack–Lehane score	1 (1–2)[1–1]	1 (1–2)[1–1]	0.42

Data presented as median (range) [25–75 interquartile range]. * statistically significant.

**Table 3 diagnostics-10-00139-t003:** Questionnaire assessing the feedback of participants.

Parameter	Non-Channeled Blade	Channeled Blade
*p*		
Quality of visualization	2 (1–4) [1–2]	2 (1–4) [1–2]
0.47		
Ease of intubation	3 (1–5) [2–4]	1 (1–4) [1–2]
<0.001 *		

Data presented as median (range) [25–75 interquartile range]. * statistically significant.

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
