# Peer review of "Video Laryngoscopic Intubation Using the King VisionTM Laryngoscope in a Simulated Cervical Spine Trauma: A Comparison Between Non-Channeled and Channeled Disposable Blades"

_diagnostics, 2020, doi:10.3390/diagnostics10030139_

Round 1

Reviewer 1 Report

This study did a comparison between non-channeled and channeled disposable blades of Video laryngoscopic intubation using the KingVision laryngoscope in a simulated cervical spine trauma. Overall, the study was well-design and the manuscript was well-written. I just have several minor suggestions.

Please add some discussion about the difference between  between non-channeled and channeled disposable blades of Video laryngoscopic intubation. Please move table 1 to supplement material Please delete figure 2 and described in text only.

Author Response

Reviewer No.1:

We would like to thank the reviewer for his/her valuable comments and suggestions.

This study did a comparison between non-channeled and channeled disposable blades of Video laryngoscopic intubation using the KingVision laryngoscope in a simulated cervical spine trauma. Overall, the study was well-design and the manuscript was well-written. I just have several minor suggestions.

Please add some discussion about the difference between  between non-channeled and channeled disposable blades of Video laryngoscopic intubation.

A: Following paragraph has been added to Discussion section:

Videolaryngoscopy blades may be divided into non-channeled and channeled. Each of the blades has its advantages and disadvantages [4,5]. Non-channeled blades are generally thinner, easier to insert and provide a good quality view to the vocal cords even in significantly limited mouth opening. Their main disadvantage is that insertion of the tracheal tube needs special angulated introducer and there may be a risk of trauma to the oropharyngeal soft tissues caused by blind spots during tracheal tube insertion. The channeled blades provide a reliable direct guide for tracheal tube insertion but may be more difficult to introduce in limited mouth opening, intraoral swelling or bulky tongue.

Please move table 1 to supplement material

A: Table 1 has been moved to Supplement material

 Please delete figure 2 and described in text only.

A: We ask the reviewer if Figure 2 could stay in the manuscript. We feel that this figure can provide the readers illustration of the difference between the blades. If required we can modify this Figure and add the image of tracheal tubes into each blade in order to illustrate clearly the difference between the blades.  

Reviewer 2 Report

This study is a prospective, non-blind, randomized study evaluating the effectiveness of novel videolaryngoscope for orotracheal intubation.

page 1, line 40 Introduction

~ intubation is spreading from operating rooms to emergency medicine departments --> please use other word than spreading

page 1, line 42

the angle of view of the vocal cords--> needs English correction

The methods, results, and discussion seem adequate and concise. This study deserves to be published after English grammar correction.

Author Response

Reviewer No.2:

We would like to thank the reviewer for his/her valuable comments and suggestions.

This study is a prospective, non-blind, randomized study evaluating the effectiveness of novel videolaryngoscope for orotracheal intubation.

page 1, line 40 Introduction

~ intubation is spreading from operating rooms to emergency medicine departments --> please use other word than spreading

A: The sentence was changed to:

Videolaryngoscopy-guided tracheal intubation is gradually moving from operating rooms to emergency medicine departments and to the out-of-hospital setting.

page 1, line 42

the angle of view of the vocal cords--> needs English correction

A: The sentence was changed to:

Videolaryngoscopy may offer some advantages against Macintosh laryngoscopy:  improved and easier visibility of the laryngeal inlet in normal and difficult laryngoscopy scenarios, using less force during intubation attempts, and visualization of the intubation process on the monitor.

The methods, results, and discussion seem adequate and concise. This study deserves to be published after English grammar correction.